# Quantitative Analysis of Velopharyngeal Movement by Applying Principal Component Analysis to Range Images Produced by a Three-Dimensional Endoscope

**DOI:** 10.3390/dj5020014

**Published:** 2017-03-23

**Authors:** Asuka Nakano, Katsuaki Mishima, Mami Shiraishi, Hirotsugu Umeda, Hiroyuki Nakano, Yoshiya Ueyama

**Affiliations:** 1Department of Oral and Maxillofacial Surgery, Graduate School of Medicine, Yamaguchi University, Minami-kogushi 1-1-1, Ube City, Yamaguchi 755-8505, Japan; nakanoasuka420@gmail.com (A.N.); smami@yamaguchi-u.ac.jp (M.S.); umeda69@yamaguchi-u.ac.jp (H.U.); uyoshiya@yamaguchi-u.ac.jp (Y.U.); 2Division of Oral and Maxillofacial Surgery, Department of Maxillofacial Diagnostic and Surgical Sciences, Kyushu University Hospital, Fukuoka 812-8582, Japan; nakano-h@dent.kyushu-u.ac.jp

**Keywords:** 3D endoscope, velopharyngeal motion, range image, principal component analysis

## Abstract

**Objectives:** The purpose of this study was to develop a new technique for analyzing velopharyngeal movement and to investigate its utility. **Materials and Methods:** Velopharyngeal motion of 20 normal individuals was analyzed. A three-dimensional (3D) endoscope was inserted into the oral cavity, and the movement of the soft palate was measured using an exclusive fixation device. Range images of the soft palate were produced during phonation of the Japanese vowel /a/, and virtual grids were then overlaid on these images. Principal component analyses were applied to the 3D coordinates of the intersections of the virtual grids. The centers of gravity of the virtual grids were calculated, and the magnitude of the shift of the grid intersections during phonation was calculated. **Results:** The first and the second principal component scores were responsible for the upper posterior direction and the upper direction, respectively. The average magnitude of the shift of the center of gravity was 4.75 mm in males and 4.33 mm in females. **Conclusions:** Quantitative analysis of velopharyngeal movement was achieved by a method of applying principal component analysis (PCA) to the range images obtained from a 3D endoscope. There was no sex difference in velopharyngeal movement.

## 1. Introduction

For individuals with a cleft palate, it is important to acquire velopharyngeal function by surgical reconstruction and postoperative speech therapy in order to develop normal speech [1,2,3]. Therefore, the precise evaluation of velopharyngeal movement is necessary in speech therapy. Although nasopharyngoscopy has been widely used to diagnose velopharyngeal closure, it can only provide a qualitative assessment [4,5]. To be able to evaluate velopharyngeal function in detail, it is necessary to measure velopharyngeal movement quantitatively.

Velopharyngeal movement is quick and complicated. This means that, to measure velopharyngeal movement quantitatively and precisely, it is necessary for us to be able to use four-dimensional information in detail, adding the temporal axis element to the three-dimensional spatial information. To illustrate various aspects of velopharyngeal movement, we have been developing techniques to measure movement of the velopharynx in detail, and we have also devised methods of producing range images of the velopharynx using a commercially available three-dimensional (3D) endoscope [6]. We have succeeded in obtaining four-dimensional (4D) velopharyngeal information by producing velopharyngeal range images using a 3D endoscope. Our previous study demonstrated that this system provides accurate and reliable range images of the velopharynx when setting the tip of the endoscope at 2–6 cm from an object by using a fixation device. The standard error of the coordinate value produced from 30 frames was within 0.2 mm three-dimensionally [7].

Because it is hard to identify the anatomical landmarks in the velopharynx, we cannot fully understand the characteristics of velopharyngeal movement based only on the magnitudes and directions of movement of arbitrary points. Furthermore, to understand the enormous amount of data obtained by the 3D endoscope, it is necessary to identify the parameters characterizing velopharyngeal movement. In other words, a parameter to extract the precise characteristics of the form change of the soft palate along the temporal axis is necessary.

Some studies have reported the ability to capture the characteristics of lip movement well by applying principal component analysis (PCA) to 4D labial information [8,9,10]. We therefore considered that this technique would also be useful for analyzing the enormous amount of data for velopharyngeal movement. The purpose of this study was to determine whether PCA was capable of characterizing velopharyngeal movement properly using this new technique.

## 2. Methods

### 2.1. Production of Range Images

The details of the endoscopic system were described in our previous article [6]. This endoscopic system (Shinko Optical Co. Ltd., Tokyo, Japan) consists of a 3D endoscope (3D-132D; Figure 1), a camera control unit (3D-SP-2001), a 3D converter (SK-1057-3D-A), and a light source (CL-75X-II). A pattern projection system was integrated into the commercially available endoscope with two 1/10-inch micro CCD cameras mounted on the tip (Figure 1). The light source was connected to the pattern, and the pattern was sent to the tip through optical fibers. Right and left camera images of the endoscope with the projected pattern were captured via an IEEE1394 FireWire cable at a sampling rate of 30 frames per second (fps) after passing through an analog to digital converter (Canopus ADVC-110; Grass Valley Co. Ltd., Kobe, Japan) connected to a workstation (Dell Precision T7500, CPU: Xenon 2.40 GHz, 6 core; Round Rock, TX, USA). The 3D converter synchronized the right and left cameras of the endoscope. The video files were separated into right and left camera images (640 × 240 pixels for each camera video file), and interlace interpolation was then performed using linear interpolation. Using the Tsai algorithm [11], the distortion of the camera images was corrected. After the camera images were rectified, the right and left images were stereo-matched and disparities were calculated using the Birchfield technique [12].

### 2.2. Analyses of Velopharyngeal Motion

Velar motions in 20 healthy adults (ten males; 25–38 years of age, ten females; 25–40 years of age) during phonation of the Japanese vowel /a/ were analyzed. To fix the head position, the volunteers were asked to sit in a chair and rest their head against a wall. The endoscope was inserted into the oral cavity and then secured using a fixation device (Figure 2). The fixation device consisted of a commercially available light-stand and a multi-jointed arm which allowed for arbitrary positioning. The continuous velopharyngeal movement during phonation of the Japanese vowel /a/ was recorded through the oral cavity, and range images of the soft palate were produced for 30 consecutive fps. Because obvious anatomical landmarks were not present, we determined a quadrangle on the soft palate below the hard palate, above the uvula, not including the lateral pharyngeal wall. The quadrangle was equally divided into a 5 × 5 virtual grid (Figure 3) [13], and principal component analyses were then applied to the 3D coordinates of the grid’s 36 intersections. Furthermore, the center of gravity of the virtual grid and the magnitude of shift during phonation of /a/ were calculated, and the differences between males and females were then statistically analyzed using a Mann–Whitney U-test (IBM SPSS ver.22, CHI, IL, USA).

### 2.3. Principal Component Analyses

Principal component analyses (PCAs) were applied to the 3D coordinates of the grid intersections. The program was based on the Mouth Motion PCATM (Ergovision, Osaka, Japan) and Body Shape BrowserTM (BSB; Ergovision, Osaka, Japan). Using this system, the first to 12th principal component scores (PCSs) were calculated, and the results were visualized. The first and second scores were plotted (first on the *x*-axis and second on the *y*-axis), and discriminant analysis was then conducted to compare the differences in the velopharyngeal movements of males and females.

This research was approved by the Institutional Review Board of Yamaguchi University Hospital, and informed consent was obtained.

## 3. Results

The average maximum magnitude of the shift of the center of gravity was 4.74 mm for males and 4.66 mm for females (*p* = 0.81). As a result of having visualized the results of the PCA using BSB, the movement in the posterior upper direction and the movement in the upper direction were found to be the first and the second PCSs in both males and females, respectively (Figure 4 and Figure 5). Graphs plotting the first and second PCSs on the *x*-axis and *y*-axis are shown in Figure 6. The first PCS showed a wider distribution in females than in males (Figure 6). Discriminant analysis was able to correctly discriminate the velopharyngeal movement for males and females in 54.6% of the cases.

## 4. Discussion

Various techniques for diagnosing velopharyngeal movement have been used including nasopharyngoscopy, computed tomography (CT), magnetic resonance imaging (MRI), cephalometric radiographic analysis, and videofluoroscopy [14]. Nasonendoscopy has also been widely used, but evaluation of velopharyngeal incompetence using an endoscope can only provide a qualitative assessment [4,5]. Therefore, at present, detailed analysis and objective evaluation of velopharyngeal movement is not possible. Because of concerns related to radiation exposure, imaging studies with irradiation are not appropriate for frequent testing. Recently, the quantitative evaluation of velopharyngeal function using MRI has been investigated [15,16]. Particularly, the utility of a quantitative analysis method of velopharyngeal motion using dynamic MRI has been reported [17]. However, MRI is distorted if speech appliances are worn, and it is not a suitable method for obtaining real-time feedback. An endoscopic measuring system was recently developed in which a pattern projection system was incorporated into a commercially available 3D endoscope [6]. Using this system, we have been developing techniques to measure movements of the velopharynx in detail, and we have also devised methods to produce range images of the velopharynx using a 3D endoscope [6].

In our current study, using an endoscopic measurement system, only measurement from the oral cavity could be made because of the large diameter of the tip of the scope. Consequently, the target for measurement was limited to a vowel sound which was pronounced with mouth opening. Furthermore, the target for measurement was limited to the soft palate because measurements could not be made over a wide area in a time.

The main purpose of PCA is to reduce the dimensionality of a data set consisting of a large number of interrelated variables, while retaining as much of the variation that is present in the data set as possible [18]. In other words, it is hoped that PCA can be used to extract important characteristics from huge data sets. The first principal component is the score having the highest level of explained variance [17]. In the present analysis, PCA applied to 3D velopharyngeal movement during phonation of the Japanese vowel /a/ revealed that the first and second PCSs were the parameters representing the movement in the posterior upper direction and the upper direction, respectively. The first and the second PCSs are thought to be major components in the PCA, and when discriminant analysis was performed with these components, no significant sex difference was found in velopharyngeal movement. However, since the sample size was small, further studies on larger samples are needed to confirm these findings and to make meaningful generalizations.

In the future, we plan to apply this technology to a fiber scope, as we think that this would allow measurement from the nasal cavity. Furthermore, we think the present technique would be appropriate for the quantification of movement at other sites due to its ability to capture the movement characteristics of a curved or irregular-shaped surface where landmark identification is difficult.

## 5. Conclusions

Quantitative analysis of velopharyngeal movement was enabled by a method of applying PCA to the range images obtained from a 3D endoscope. PCA is a technique that captured the characteristics of velopharyngeal movement. Discriminant analysis showed no clear sex difference in velopharyngeal movement.

## Figures and Tables

**Figure 1 dentistry-05-00014-f001:**
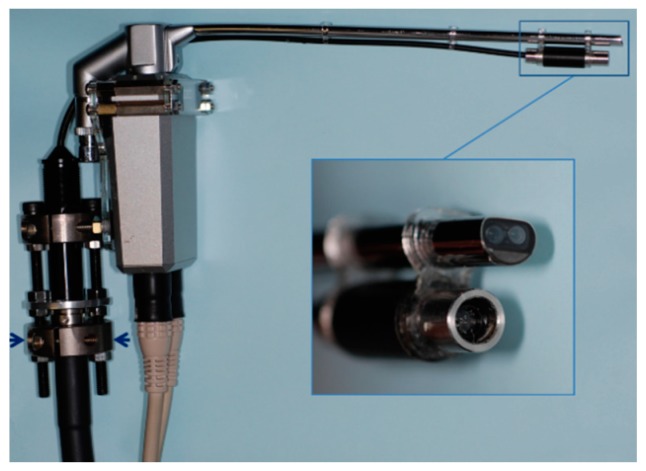
The endoscope was 5.4 mm in diameter. Two 1/10 inch micro charge-coupled device cameras are mounted into the tip of the endoscope. The lens of the pattern projection system, which is mounted in the system’s tip. A latticework pattern is placed on the bottom of the endoscope (arrow).

**Figure 2 dentistry-05-00014-f002:**
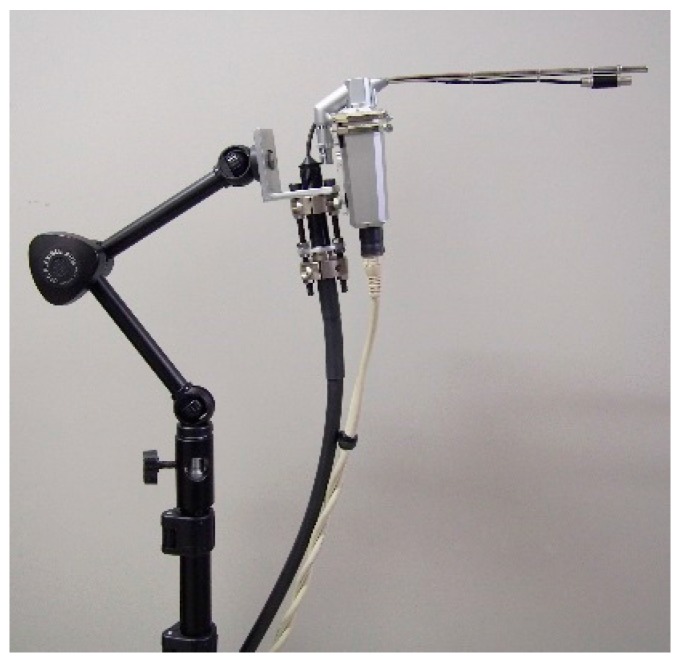
Fixation device comprising a commercially available light stand and a multi-jointed arm. This system enables arbitrary positioning.

**Figure 3 dentistry-05-00014-f003:**
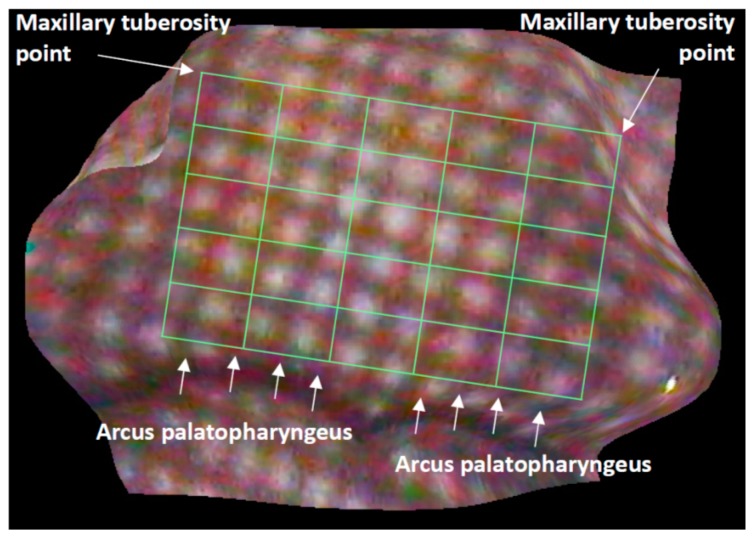
Production of a virtual grid. First, a quadrangle is determined on the soft palate below the hard palate, above the arcus palatopharyngeus, excluding the lateral pharyngeal wall, and then the quadrangle is divided equally into a 5 × 5 grid.

**Figure 4 dentistry-05-00014-f004:**
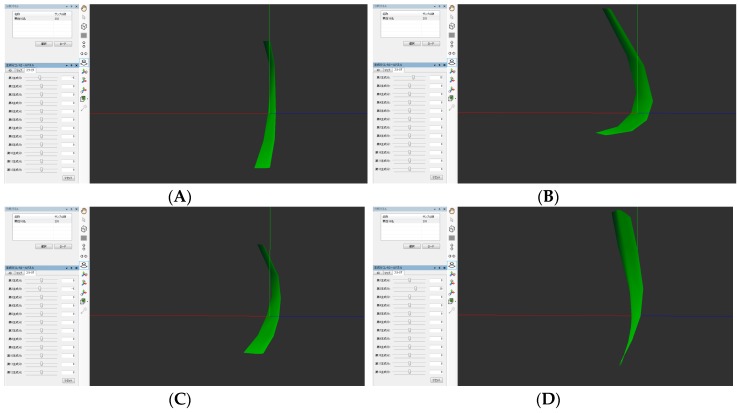
The arbitrarily determined quadrangle shapes on the soft palate observed from the lateral direction in a male are shown. The soft palate shapes corresponding to the first PCSs are shown in Figure 4 (**A**,**B**), and the second PCSs are shown in Figure 4 (**C**,**D**). The movement in the posterior upper direction was found at the first PCSs and the movement in the upper direction was found at the second PCSs.

**Figure 5 dentistry-05-00014-f005:**
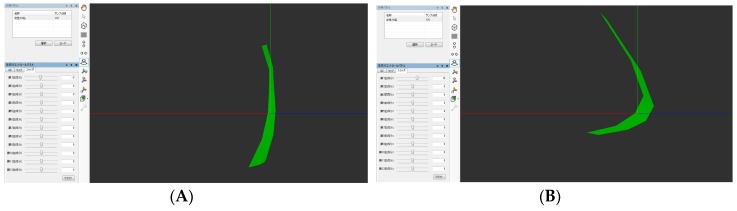
The quadrangle shapes arbitrarily determined on the soft palate observed from the lateral direction in a female are shown. The soft palate shapes corresponding to the first PCSs are shown in Figure 5 (**A**,**B**), and the second PCSs are shown in Figure 5 (**C**,**D**). The movement in the posterior upper direction was found at the first PCSs and the movement in the upper direction was found at the second PCSs.

**Figure 6 dentistry-05-00014-f006:**
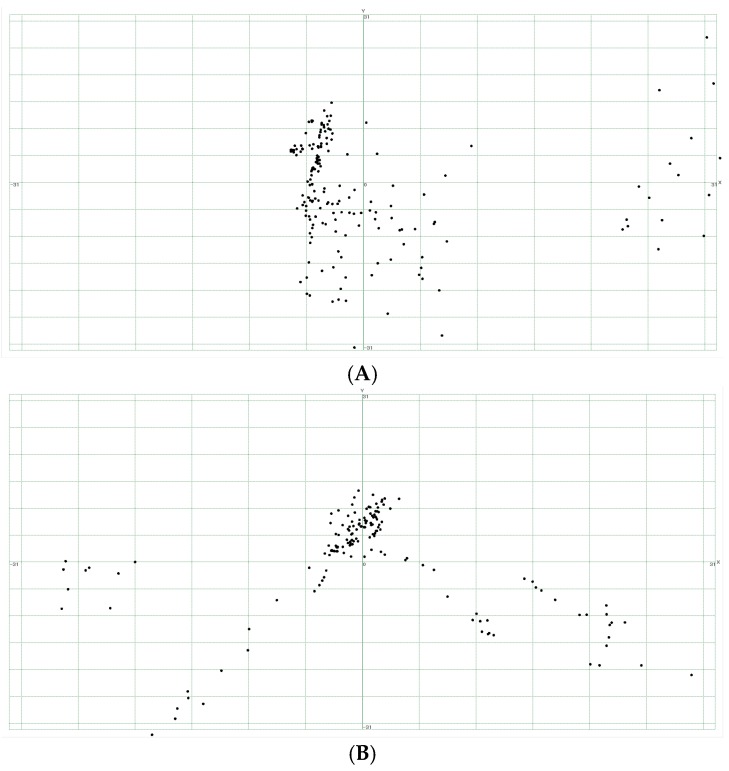
Graphs plotting the first and second PCSs on the *x*- and *y*-axes are shown. Graphs plotting the first and second PCSs in a male are shown in (**A**) and in a female, in (**B**).

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
