# Peer review of "Quantitative Analysis of Velopharyngeal Movement by Applying Principal Component Analysis to Range Images Produced by a Three-Dimensional Endoscope"

_dentistry, 2017, doi:10.3390/dj5020014_

Round 1

Reviewer 1 Report

A brief summary (one short paragraph) outlining the aim of the paper and its main contributions.

The aim of this paper is to demonstrate how Principal Components Analysis (PCA) can be applied to analyze velar movement using nasendoscopy.  A sample size of n=20 underwent nasendoscopy, a 5x5 quadrangle was projected onto the velum, and PCA was used to determine which quadrants contributed most to velar elevation.  Upper and posterior upper quadrants were found to move the most.  This research is innovative in that it aims to quantify visualization of the velum from nasendoscopy, which has traditionally provided only qualitative information. 

Broad comments highlighting areas of strength and weakness.

Very interesting study that has merit to the field.  There are three major concerns with this study.  First, it is unclear how you positioned the endoscope in each participant to prevent angle deviations between the camera and the velum.  Deviations in the angle of the camera and/or velum would change the locations of the virtual grid on the surface of the velum.  Secondly, the descriptions of your locations (e.g. posterior upper and upper) are ambiguous and difficult to understand.  An added figure showing the location and movement of these may help readers understand.  Lastly, I’m concerned that the statistical assumptions for performing a PCA have not been met with this study for the following reasons: no mention of linearity and presence of outliers.  PCA section of article should include mention of assumptions and verification for methods chosen, as this paper revolves around the PCA. There is too little information provided.

Specific comments referring to line numbers, tables or figures.

Abstract: Add a sentence to the results/conclusion section that provides a statement to the broad overall impact of the study that helps the readers understand how results are related to clinical impact

Introduction:

35: researchers have argued that nasopharyngoscopy can also provide quantitative data, please find these articles and amend text. While I agree this is still largely up for qualitative interpretation, not all will agree with your statement here.  

36:  There have been several MRI studies on the velopharynx that look at velopharyngeal movement 3-dimensionally, including dynamic MRI/MRI movies.  Rahimian et al., 2016 addressed track and predicting velar and pharyngeal wall locations during dynamic MRI sequences, for example. Consider providing a more comprehensive review of these studies.

47: please explain in more detail the fixation device—does it ensure the right and left scopes are at the exact same distance from the portal and how is this determined. It is well known that in stereoscopy and use of 2 cameras the depth cues are dependent upon depth from the object relative to distance between cameras. Perhaps your methods clearly account for this, however, your description should be improved to help the reader understand this prior published method that you describe—particularly given this method is the basis for the present study expansion.

52: explain in your introduction which variables you feel are the ones that you feel are important to be captured and quantified.

56: “…was to determine whether it was… specify what “it” is. Sentence could be improved, it is awkward as written.

Methods:

2.1: again here you should explain how you ensured equal distance from portal and camera tip between both scopes and how you ensured the scopes remained at the same distance and were not moving throughout the procedure—thus greatly effecting the depth cues. Perhaps an image of this set up on a patient may be helpful. Although the reader can reference the original work, it will greatly improve the readability of the present study.

2.2 can you specify what you measuring or viewing in regards to velopharyngeal motions---given ths is a scope it is important for readers to know the view and what “motions” you are speaking of. There are limitations to what can be seen because it is a scope and so the general term VP motions is a bit misleading (see later comment for study limitations).

80: here you could reference the image provided to display the device.

Results/Discussion/Figures:

Limitations—add information about the study limitation related to scopes being invasive, providing only particular view points (explain this & emphasize what cannot be seen or realized from scopes), and less tolerated by children. And so this method proposed is really only for gaining insights for research purposes and may need much developments before offering clinical value. Also, for research, it it unlikely this will be tolerated in children and this should be highlighted.

125:  Your statement is unclear, are you referring to diagnosing dysfunction or quantifying movement; I assume you don’t mean diagnose VP movement but maybe describe VP motions? Imaging methods do not “diagnose” movement.

128: what is meant by they are not yet accessible? Those techniques are all commonly used and some (e.g., MRI) have been shown to have excellent ability to quantify VP motions and perhaps with even more of a comprehensive approach because they look at the whole system and provide insight into muscle function. This section neglects to highlight the limitations known regarding scopes and particular the methods described here (see comment above). It may be advisable to provide a better balance to reporting of information and show less bias regarding methods. It is definitely true that the real-time feedback is the limitation of other methods. But there are significant limitations to scope that are not mentioned here.  

154: Discriminant analysis is different from PCA, which was performed?

Figure 1 is not very helpful. Can you provide anatomic landmarks to reference the grid

Figures 2 & 3, it is not possible to read the text to the left of the images. You should crop the image to remove the computer backdrop and all information that is not relevant.

It seems what is missing is discussion of future implications of these developments. Is there any potential clinical usefulness of these methods and if so, describe what insights this could provide. The study focuses more on the sex differences. What other purposes could this approach be useful for? Although this is more of a methods report, you may gain more interest from readership if you can explain the potential implications of findings.

Author Response

First, it is unclear how you positioned the endoscope in each participant to prevent angle deviations between the camera and the velum. Deviations in the angle of the camera and/or velum would change the locations of the virtual grid on the surface of the velum.

->>        We measured it while almost keeping the angle between the soft palate and the scope fixed, but in the first place the purpose that set a virtual grid was to cancel the error in measurement.

Secondly, the descriptions of your locations (e.g. posterior upper and upper) are ambiguous and difficult to understand.

->>        The following description was added.

The volunteers were asked to sit in a chair and rest their head against a wall to fix the head position. The endoscope was “inserted into the oral cavity and” then secured using the fixation device.

Lastly, I’m concerned that the statistical assumptions for performing a PCA have not been met with this study for the following reasons: no mention of linearity and presence of outliers.

->>        Sorry, unfortunately, linearity and presence of outliers were not investigated.

47: please explain in more detail the fixation device—does it ensure the right and left scopes are at the exact same distance from the portal and how is this determined. It is well known that in stereoscopy and use of 2 cameras the depth cues are dependent upon depth from the object relative to distance between cameras. Perhaps your methods clearly account for this, however, your description should be improved to help the reader understand this prior published method that you describe—particularly given this method is the basis for the present study expansion.

->>        The photograph of the fixation device was added (Figure 2).

The endoscope used was a rigid scope that two CCD cameras were mounted into the tip of the endoscope. Therefore, a position and the posture between the two cameras did not change.

52: Explain in your introduction which variables you feel are the ones that you feel are important to be captured and quantified.

->>        The following sentences were also added in the Introduction section.

“In other words, a parameter to extract the characteristic of the form change of the soft palate along temporal axes precisely is necessary.”

56: “…was to determine whether it was… specify what “it” is. Sentence could be improved, it is awkward as written.

->>        "It" means PCA.

2.2 can you specify what you measuring or viewing in regards to velopharyngeal motions given ths is a scope it is important for readers to know the view and what “motions” you are speaking of. There are limitations to what can be seen because it is a scope and so the general term VP motions is a bit misleading.

->>        The range that we can measure by the endoscope was limited considerably.

              This description was added in the Discussion section.

80: here you could reference the image provided to display the device.

->>        The photograph of the fixation device was added (Figure 2).

Limitations

->>        Because we measured it from the oral cavity, we could apply it to children.

              However, the following sentences about “limitation” was added in the Discussion section.

“In the current our endoscopic measurement system, only measurement from the oral cavity could be made because the tip diameter of the scope was big. Therefore, the target for measurement was limited to a vowel sound which was pronounced with mouth opening. Furthermore, the target for measurement was limited to the soft palate because a wide range could not be measured in a time.”

36: There have been several MRI studies on the velopharynx that look at velopharyngeal movement 3dimensionally, including dynamic MRI/MRI movies. Rahimian et al., 2016 addressed track and predicting velar and pharyngeal wall locations during dynamic MRI sequences, for example. Consider providing a more comprehensive review of these studies.

128: what is meant by they are not yet accessible? Those techniques are all commonly used and some (e.g., MRI) have been shown to have excellent ability to quantify VP motions and perhaps with even more of a comprehensive approach because they look at the whole system and provide insight into muscle function.

->>        The following sentences was added in the Discussion section. “Particularly, the utility of the quantitatively analyzing method of the velopharyngeal motion using the dynamic MRI has been reported [17].”

154: Discriminant analysis is different from PCA, which was performed?

->>        We performed discriminant analysis to determine whether we can detect gender difference using PCS.

Figure 1 is not very helpful. Can you provide anatomic landmarks to reference the grid

->>        The anatomical landmarks were added in the figure.

Figures 2 & 3, it is not possible to read the text to the left of the images. You should crop the image to remove the computer backdrop and all information that is not relevant.

->>        These figures were revised.

It seems what is missing is discussion of future implications of these developments.

->>        In the future, we plan to apply this technology to a fiber scope. We think that the measurement from the nasal cavity is thereby enabled. In addition, we think the present techniques to be available for quantification of the movement at the other sites because we can catch a characteristic of the movement of the curved surface of the complicated shape that is difficult to extract landmarks.

Reviewer 2 Report

Velopharyngeal movements during speech have always been an interesting topic. Different imaging procedures have been studied for assessing the velopharyngeal sphincter anatomy and physiology during speech and swallowing including videonasopharyngoscopy, multiplanar videofluoroscopy, ultrasound imaging and magnetic resonance imaging. To date the best approach for a clinical evaluation of velopharyngeal insufficiency is the combination of videonasopharyngoscopy and videofluoroscopy. The main disadvantages of endoscopy for assessing velopharyngeal movements during speech are the impossibility of doing actual size measurements and the lack of 3-D conceptualization of the sphincter in motion. The paper describes a new methodology for using endoscopy and providing 3-D imaging and measurement. The technology presented is definitely promising. However, the authors used a very limited speech sample (a single vowel sound or phoneme : /a/). It is necessary to describe in detail what are the characteristics of the vocal tract during the production of the phoneme used for the study. It is also necessary to address the issue that velopharyngeal motion during vowel sounds is limited. Velopharyngeal seal occurs during consonant sounds with increased intraoral pressure regardless of the language in question. The authors should explain why they used only a vowel sound and the severe limitation of this selection. It is also necessary to elaborate on movements of other components of the velopharyngeal sphincter, not only the velum but the lateral pharyngeal walls and the posterior pharyngeal wall. The study does not address the presence - absence of Passavant's ridge at all.

The paper can be published if the authors address the queries mentioned herein and delineate the limitations of their study and the need for future and extensive research of the methodology described in the paper

Author Response

1. The authors should explain why they used only a vowel sound and the severe limitation of this selection.

->>        The following sentences were added in the Discussion section.

“In the current our endoscopic measurement system, only measurement from the oral cavity could be made because the tip diameter of the scope was big. Therefore, the target for measurement was limited to a vowel sound which was pronounced with mouth opening.”

2. It is also necessary to elaborate on movements of other components of the velopharyngeal sphincter, not only the velum but the lateral pharyngeal walls and the posterior pharyngeal wall. The study does not address the presence absence of Passavant's ridge at all.

->>        The following sentences were also added in the Discussion section. “Because a wide range could not be measured in a time, the target for measurement was limited to the soft palate.”

Round 2

Reviewer 1 Report

Although some improvements have been made the following prior suggestions have not been considered or incorporated into the paper.

Prior comment not addressed:

Abstract: Add a sentence to the results/conclusion section that provides a statement to the broad overall impact of the study that helps the readers understand how results are related to clinical impact

Introduction: researchers have argued that nasopharyngoscopy can also provide quantitative data, please find these articles and amend text. While I agree this is still largely up for qualitative interpretation, not all will agree with your statement here. Specifically these are articles that report methods that quantitative data can be obtained from scope data.

Results: what is meant by they are not yet accessible? Those techniques are all commonly used and some (e.g., MRI) have been shown to have excellent ability to quantify VP motions and perhaps with even more of a comprehensive approach because they look at the whole system and provide insight into muscle function. This section neglects to highlight the limitations known regarding scopes and particular the methods described here (see comment above). It may be advisable to provide a better balance to reporting of information and show less bias regarding methods. It is definitely true that the real-time feedback is the limitation of other methods. But there are significant limitations to scope that are not mentioned here.  -to me this has not been appropriately responded to.

New concern: it was new information that the scope was positioned orally, which is not routinely used when scoping for velopharyngeal assessments. This is a significant impacting factor that should be made clear by mentioning this in the abstract and when talking about "velar motions" that it is really oral side velar motions. Some will argue this significantly decreases the value of your findings.

Author Response

1.    Add a sentence to the results/conclusion section that provides a statement to the broad overall impact of the study that helps the readers understand how results are related to clinical impact.

->>        The following sentences were added in the last of Abstract section. “Results: The first and the second principal component scores were responsible for the upper posterior direction and the upper direction, respectively. The average magnitude of the shift of the center of gravity was 4.75 mm in males and 4.33 mm in females. Conclusions: Quantitative analysis of velopharyngeal movement was achieved by a method of applying PCA to the range images obtained from a 3D endoscope. There was no sex difference in velopharyngeal movement.”

2.    Introduction: researchers have argued that nasopharyngoscopy can also provide quantitative data, please find these articles and amend text.

->>        P.1, L.33-35; We have described as follows: “Although nasopharyngoscopy has been widely used to diagnose velopharyngeal closure, it can only provide qualitative assessment [4, 5].” In other words, we did not state that nasopharyngoscopy can provide quantitative data.

3.    Results: what is meant by they are not yet accessible? Those techniques are all commonly used and some (e.g., MRI) have been shown to have excellent ability to quantify VP motions and perhaps with even more of a comprehensive approach because they look at the whole system and provide insight into muscle function. This section neglects to highlight the limitations known regarding scopes and particular the methods described here (see comment above). It may be advisable to provide a better balance to reporting of information and show less bias regarding methods. It is definitely true that the realtime feedback is the limitation of other methods. But there are significant limitations to scope that are not mentioned here. To me this has not been appropriately responded to.

->>        The following sentence was removed. “, but they are not yet accessible” (P. 5, L.136).

There is the fault that “the target for measurement was limited to the soft palate because measurements could not be made over a wide area in a time”. This mention has already described it in P.6, L.151.

We think that the reviewer described a section of Discussion, not a section of Results.

4.    New concern: it was new information that the scope was positioned orally, which is not routinely used when scoping for velopharyngeal assessments. This is a significant impacting factor that should be made clear by mentioning this in the abstract and when talking about "velar motions" that it is really oral side velar motions. Some will argue this significantly decreases the value of your findings.

->>        The following sentence was added in Abstract section. “A three-dimensional (3D) endoscope was inserted into the oral cavity, and the movement of the soft palate was measured using an exclusive fixation device.”

5.    This second revised manuscript has undergone English editing again.

Round 3

Reviewer 1 Report

Reviewers have appropriately addressed the concerns from the prior review.